# The Role of Gut Microbiota in the Clinical Outcome of Septic Patients: State of the Art and Future Perspectives

**DOI:** 10.3390/ijms24119307

**Published:** 2023-05-26

**Authors:** Nadia Marascio, Giuseppe Guido Maria Scarlata, Francesco Romeo, Claudia Cicino, Enrico Maria Trecarichi, Angela Quirino, Carlo Torti, Giovanni Matera, Alessandro Russo

**Affiliations:** 1Clinical Microbiology Unit, Department of Health Sciences, “Magna Graecia” University of Catanzaro, “Mater Domini” Teaching Hospital, 88100 Catanzaro, Italy; nmarascio@unicz.it (N.M.); giuseppescarlata93@gmail.com (G.G.M.S.); claudiacicino@gmail.com (C.C.); quirino@unicz.it (A.Q.); mmatera@unicz.it (G.M.); 2Infectious and Tropical Diseases Unit, Department of Medical and Surgical Sciences, “Magna Graecia” University, “Mater Domini” Teaching Hospital, 88100 Catanzaro, Italy; francy.romeo95@gmail.com (F.R.); em.trecarichi@unicz.it (E.M.T.); torti@unicz.it (C.T.)

**Keywords:** gut microbiota, sepsis, next-generation sequencing (NGS), clinical outcome

## Abstract

Sepsis is a life-threatening multiple-organ dysfunction caused by a dysregulated host response to infection, with high mortality worldwide; 11 million deaths per year are attributable to sepsis in high-income countries. Several research groups have reported that septic patients display a dysbiotic gut microbiota, often related to high mortality. Based on current knowledge, in this narrative review, we revised original articles, clinical trials, and pilot studies to evaluate the beneficial effect of gut microbiota manipulation in clinical practice, starting from an early diagnosis of sepsis and an in-depth analysis of gut microbiota.

## 1. Introduction

The gut microbiota is composed of 35,000 different species; among these, *Bacteroides* and *Firmicutes* make up more than 90% of the phylogenetic categories found in the human gut [1]. The microbiota is involved in nutrient metabolism, immune modulation, and protection of the gastro-intestinal tract. The primary source of energy is dietary carbohydrates, and fermentation of carbohydrates and oligosaccharides by colon microorganisms such as *Bifidobacterium* and *Fecalibacterium* results in the synthesis of short-chain fatty acids such as butyrate and acetate, which are rich sources of energy for the host [2,3,4]. The gut microbiota has also been shown to have a positive impact on lipid metabolism through suppressing the inhibition of lipoprotein lipase activity in adipocytes and increasing the efficiency of lipid hydrolysis [5].

Of importance, the microbiota is also involved in protein metabolism. Specifically, several amino acid transporters on the bacterial cell wall facilitate the entry of amino acids from the intestinal lumen into the bacteria, where different gene products convert amino acids into small signaling molecules and antimicrobial peptides [6]. It is well known that the venous system of portal circulation represents the anatomical base of multiple functional interactions between the gastrointestinal tract and the liver. This axis seems to be an important feature that protects the host against potentially harmful and toxic substances from the gut, thereby maintaining the homeostasis of the immune system [7]. Crosstalk in the gut and liver axis is widely acknowledged, as the gut and liver communicate bidirectionally via biliary, portal, and systemic circulation [8].

Gut dysbiosis can be defined as the imbalance of gut microbiota [9] associated with an unhealthy outcome and with infectious diseases [10,11]. Several research groups have reported that septic patients display a dysbiotic gut microbiota [12]. Sepsis is a life-threatening multiple-organ dysfunction caused by a dysregulated host response to infection, with high mortality worldwide. According to the Sepsis-3 definition, a new algorithm involving both the Sequential Organ Failure Assessment (SOFA) and quick-SOFA scores allows a homogeneous identification of septic patients [13,14]. Furthermore, 11 million deaths per year are attributable to sepsis in high-income countries [15]. Based on current knowledge, in this narrative review, we revised original articles, clinical trials, and pilot studies to evaluate beneficial effect of gut microbiota manipulation in clinical practice, starting from an early diagnosis of sepsis and an in-depth analysis of gut microbiota. As a matter of fact, a targeted therapy of gut microbiota may be a promising strategy for sepsis. Here, we discuss the role of intestinal flora in the development of sepsis and summarize the latest progress in the field of targeted treatment of sepsis.

## 2. Timely Diagnosis of Sepsis and the Role of Fast Microbiology

Many studies highlight the need for a clearer understanding of the pathophysiological mechanisms underlying sepsis to achieve an early and timely diagnosis and to set up strategic therapies [16]. The dysregulated host response to infection leading to sepsis and septic shock is a life-threatening event that, despite advances in organ support and antimicrobial therapy, is associated with a high mortality rate [17]. Despite implementation of international guidelines supporting early goal-directed therapy, recent randomized trials have demonstrated that these interventions do not improve survival of septic patients [11].

This evidence warrants an urgent clarification of the molecular mechanisms underlying clinical response in patients with sepsis or septic shock. The key to improving these processes lies in acquiring in-depth knowledge of the intricate interplay between host defense, infection, and pathogen virulence as well as the timing and types of interventions that are most effective according to the personal characteristics of individual patients. Of importance, the pharmacokinetic and pharmacodynamic properties of antibiotics should be considered because of changes in clearance and volume of distribution that are frequently observed in critically ill patients, with the potential to influence concentration of the drug at sites of infection [11].

In recent years, an increased frequency of Gram-negative MDR pathogens such as MDR *Acinetobacter baumannii* (MDR-AB) and carbapenemase-producing *Klebsiella pneumoniae* (KPC-Kp) has been observed [18,19]. It is important, therefore, to develop new diagnostic techniques to obtain a fast microbiology with early microbiological reports. These techniques can include nucleic acid amplification technologies (NAATs) that amplify nucleic acid sequences to a detectable level and identify the infecting agent or the status of the immune response. The detection of bacterial DNA fragments via real-time polymerase chain reaction (RT-PCR) in blood samples and the detection of 16S rRNA fragments of Gram-positive and Gram-negative bacteria or 18S rRNA fragments of *Candida* spp. seem to be very promising for shortening pathogen identification, as they have shown a high degree of specificity and sensitivity, thus reducing mortality, length of hospitalization, and ICU stay of patients [20]. However, these techniques are not sufficient to differentiate sepsis from other inflammatory processes, and there are no biomarkers able to identify only septic patients [11].

Currently, blood culture is still considered the “gold standard” to diagnose sepsis. However, the time to obtain the result is long, and this methodology does not meet the clinical requirements to obtain a rapid result. Other inflammation markers such as CRP (C-reactive protein) and PCT (Procalcitonin) are also used for diagnostic purposes, although they have limitations in sensitivity and specificity that are still insufficient and are affected by some diseases [21,22,23,24,25]. For the early diagnosis of this disease, important work has been carried out in recent years to find useful and accurate biomarkers. Procalcitonin is considered the best-studied biomarker for differentiating infectious from non-infectious states. In general, the markers studied are linked to inflammatory mechanisms, in the hope that they can integrate or replace the currently used ones, such as CRP and procalcitonin PCT [26]. The study by Matera et al. supports the role of PCT through showing a direct neutralizing effect on Lipopolisaccaride (LPS), suggesting a significant inhibition of the main mediators of the LPS-stimulated Th1, Treg, and monocyte activation cascade [27]. Recent data were reported about the role of presepsin as a new biomarker useful for the early diagnosis of sepsis, showing that presepsin may be used as a biomarker of sepsis in the clinical setting [28]. Figure 1 shows a flowchart about the microbiological approach to sepsis.

## 3. Gut microbiota Analysis via High-Throughput Methods

In this scenario, microbiota may have an important role in identifying patients at high-risk progression to septic status. Next-generation sequencing (NGS) approaches help to better understand the symbiotic crosstalk between the microbiota and host, which is impaired during the septic process [31]. Descriptive metagenomics investigates the microbial composition and its relative abundance in several body districts under different physiological conditions [32]. A schematic overview of gut microbiota analysis is reported in Figure 2.

The taxonomy of gut microbiota composition is performed via 16S ribosomal RNA (16SrRNA) sequencing [33]. The 16SrRNA gene contains nine hypervariable regions (V) alternated with conserved regions (Figure 3). The hypervariable regions (from V1 to V9) are the ideal target to correctly classify the bacteria [34].

The most used primer pairs to amplify genomic DNA are F27-R534, F9-R534 (comprising V1-V3), F357-R926 (V3-V5), and F515-R926 (V4-V5) [35]. The amplified V 16SrRNA region is related to the study design and sample type under investigation [36]. After amplification of the selected hypervariable regions, the resulting amplicons are sequenced [37]. Subsequently, the obtained data are processed into operational taxonomic units (OTUs). An OTU above 97% is estimated to define a species, while OTUs with sequence similarities of 95% and 80% are used to define genus and *phylum*, respectively [38]. Computational alignment using specific free databases available online, such as SILVA or Greengenes, is performed for final taxonomic identification [39,40]. Finally, alpha-diversity (observed richness relative to the number of *taxa*) and beta-diversity (the identity of observed *taxa*) among samples within a group are measured [41]. The primary source of energy is dietary carbohydrates. Fermentation of carbohydrates and oligosaccharides by colon microorganisms such as *Bacteroides*, *Bifidobacterium*, and *Fecalibacterium* leads to the synthesis of short-chain fatty acids, such as butyrate and acetate, which are a rich source of energy for the host [42]. In this regard, metabolomic analysis is usually performed on fecal and serum samples [43]. The 16SrRNA analysis on three different NGS platforms, including healthy control subjects or septic patients versus healthy controls, showed greater efficiency of sequencing V3-V4 regions for faecal microbiota taxonomy (Table 1).

Whon and coworkers evaluated the faecal microbiota of 172 South Korean patients via sequencing different regions of the 16S rRNA. Quantitative characteristics of the V1-3 dataset resulted in lower coverage of the human faecal microbiota. Alpha-diversity analyses comparing V1-3, V3-4 and V4 datasets also showed significant differences in diversity, richness, and uniformity indices. These indices in the V3-4 and V4 data sets were 2–3 times higher than those in the V1-3 data set [44]. Likewise, Abellan-Schneyder and colleagues sequenced the gut microbiota of 33 different patients. Sequence data analysis showed the identification of 9-13 *taxa* using primer pairs specific to the V3-V4 regions of 16srRNA. On the contrary, using primer pairs specific to V4-V5 regions, some genera were not detected. This study suggests how the use of primers for V3-V4 regions are largely efficient [45]. On the other hand, Kameoka and colleagues enrolled a total of 192 healthy, randomly selected Japanese volunteers to perform the sequencing of faecal samples using two different pair of primers for the V1-V2 and V3-V4 regions of 16S rRNA. Their results indicated that the bacterial composition derived from the V3-V4 region might differ from the actual abundance of specific gut bacteria and suggested the use of the V1-V2 region for Japanese population studies. This finding confirms the great heterogeneity of the metagenomic approach and that the target regions should be chosen based on the population under investigation [46]. Finally, Chen et al. compared three pairs of primers targeting the V1-V2, V3-V4, and V4 regions of the 16S rRNA in faecal samples from a small cohort of healthy subjects. The different primer pair showed different abundances of genera between the 16S rRNA regions under investigation [47]. Among septic patients, there is a lack of case–control studies that use different pairs of primer to target the V1-V2 [33] or V3-V4 16S rRNA regions [48,49,50]. Overall, further investigations of large cohorts of subjects and under different physiological and pathological conditions are needed to guide the choice of 16S rRNA regions to be sequenced [51].

## 4. Interaction between Sepsis and Gut Microbiota: Clinical Point of View

As reported above, a key role in sepsis is the highly heterogeneous syndrome caused by a disproportionate host response to infections, with subsequent depletion of immune resources and alteration of physiological homeostasis [52]. During sepsis, there is an important distortion of the gut microbiota’s composition characterized by an important reduction of microbial species diversity, a large reduction (or absence) of commensal bacterial species, and overgrowth of potential pathogen species, most commonly of *Enterococcus* spp. or *Staphylococcus* spp. [53]. Several studies underline the role of intestinal microbiota in the systemic immune response against infections, and it has been hypothesized that changes in its composition could potentially predispose the patient to a state of immunosuppression. Gut dysbiosis promotes pathogenic microbial overgrowth and translocation of intestinal pathogen-associated molecular patterns (PAMPs) to the lymphatic and portal systems, impairing the body’s defense against infection and aggravating organ damage [54]. An altered microbiota negatively affects inflammatory responses and influences the intestinal barrier’s permeability, facilitating the translocation of pathogens into systemic circulation and, consequently, increasing the risk of organ failure and sepsis. Indeed, it has been shown that during acute respiratory distress syndrome (ARDS) (very frequently associated with sepsis), the physiological pulmonary microbiota is enriched with bacteria translocated from the intestine; moreover, this increase in the number of these communities in the lungs, most frequently *Bacteroides* spp, is directly proportional to the intensity of systemic inflammation [52]. Furthermore, sepsis can progress to multiple-organ dysfunction syndrome (MODS), sustained by pro-inflammatory cytokines, such as tumor necrosis factor alpha (TNFα), interleukin-1beta (IL-1β), IL-6, IL-10, and IL-12 [55]. Nowadays, it is known that a reduction of the gut microbiota’s diversity, especially the alteration of the ratio between *Firmicutes* and *Bacteroides’* phyla, is associated with a reduced survival rate in critical patients: it has been demonstrated, through studies conducted on patients with a recent history of hospitalization for *Clostridium difficile* (known for the important structural alterations it causes to the intestinal microbiota), that dysbiosis negatively affects the rehospitalization rate for subsequent sepsis events and increases mortality in critically ill patients [50].

## 5. The Impact of Antimicrobial Therapies and MDR Pathogens on Microbiota

Some studies on laboratory rats have also shown increased mortality due to bacterial infections in mice depleted of intestinal microbes via antibiotic therapy compared to a healthy population [56]. There are also conditions, often associated with sepsis, which negatively affect the intestinal microbiota’s physiological activities and, in the same way, induce changes in the pulmonary and skin microbiota. Among these, there are invasive clinical practices necessary for the patient’s survival such as parenteral nutrition or mechanical ventilation, but also daily patient management practices and the administration of certain classes of essential drugs—above all, antibiotics, opioids, and proton pump inhibitors. During sepsis, antibiotic treatments cannot be avoided, and often, antimicrobial therapy lasts for long periods. Compelling evidence has shown that a delay in the beginning of appropriate antibiotic therapy is a risk factor for mortality; therefore, administration of antibiotics is recognized as a key component in the early treatment of sepsis [57]. It’ s now known that the use of antibiotics causes deep distortions in microbiota composition [58,59]. The usage of antibiotics leads to a loss of important taxa (with a consequent reduction of species diversity), alters the processes of certain metabolic pathways, and induces the microbiota to enter into a state of resilience against pathogens, reducing its protective potential even months after the suspension of antibiotic therapy. This is due to the direct bactericidal effect that antibiotics exert on intestinal microorganisms, particularly broad-spectrum antibiotics (Table 2).

These act on many microbial species, also influencing the metabolism of non-pathogenic organisms through imparting a selective pressure which creates the ideal conditions for the growth of enteropathogenic agents: for example, colitis from *Clostridium difficile* following long antibiotic treatments, or via promoting the growth of multi-drug resistant (MDR) bacterial species such as Vancomycin-resistant *Enterococcus* (VRE) [60]. In addition to the damage directly related with bactericidal effects, there are other alterations that occur after long-term antimicrobial therapy. This is due to co-dependency, interactions, and competition for space and nutrients between the various intestinal microbial species. Numerical reduction or alteration of some metabolic pathways of one of them can therefore involve not only the limitation, or total arrest, of beneficial bacterial colonies’ growth (not directly affected by antibiotics’ effect) but also causes the excessive growth of bacterial pathogen species due to a lack of competition in the intestinal microenvironment [61]. Different classes of antibiotics interact differently with intestinal populations, generating short- and long-term effects, even on other organs, which include conditions ranging from uncomplicated diarrhoea to life-threatening conditions [62]. Third-generation cephalosporins strongly influence Gram-negative populations, especially after repeated treatments with ceftriaxone; it has been observed that a marked depletion of *Enterobacteriaceae* colonies is often accompanied by an increase in colony numbers of *Enterococci* spp. and *Candida* spp. and an increased risk of developing *Clostridium difficile* enterocolitis [63]. It has also been demonstrated that repeated treatments with third-generation cephalosporins are associated with a greater risk of developing colonies of *Enterobacteriaceae* and, to a lesser extent, of Gram-positive species resistant to multiple classes of antibiotics [64]. The use of amoxicillin, without or in association with a beta-lactamase inhibitor, has been related to a depletion of *Lactobacillus* spp. and an increase in MDR *Enterobacteriaceae* species; these alterations affecting the intestinal microbiome can last up to 2 months [65]. Clindamycin is largely eliminated through the biliary tract and therefore has very high faecal concentrations. Its use leads to profound long-term alterations of the intestinal microbiota, characterized by a strong reduction in anaerobes compensated by a slight increase in Gram-positive and enterobacteria. It also promotes the development of MDR pathogens and increases the risk of developing colitis from *Clostridium difficile* [66]. Fluoroquinolones are also partially eliminated through the biliary tract and about 30% of the ingested dose is found in the faeces. Their use causes a strong reduction of Gram-negative aerobes not followed by a recolonization from Gram-positive bacteria or yeasts, causing a reduction of about one-third of the total taxa and an increased risk of developing quinolone-resistant species. These alterations have a highly variable duration between subjects but, generally, the gut microbiota returns to its physiological conditions within 4 weeks after the suspension of the antibiotic therapy [67]. The oral use of metronidazole in monotherapy does not particularly affect the composition of the intestinal microbiota. However, it has been noticed that an increase in the total number of *Enterococcus* spp. and macrolides resistant Gram-negative species occurred after using metronidazole in combination with clarithromycin for the treatment of *Helicobacter pylori* gastritis. On the other hand, there was a reduction of the total number of *Bifidobacteria, Clostridia,* and *Bacteroides* spp. [68]. The oral use of vancomycin is related to a reduction of *Enterococcus, Clostridia*, *Bifidobacteria,* and *Bacteroides* spp., but stimulates the growth of *Enterococcus* spp. with less susceptibility to glycopeptides and potentially pathogenic *Lactobacillaceae* and *Enterobacteriaceae* spp. [69]. Overall, there are few studies which compare the gut microbiota in septic patients to healthy control subjects. One of them was carried out in a small cohort of septic shock patients attending an Intensive Care Unit (ICU). In septic shock patients, bacterial diversity was decreased, but *Proteobacteria* and *Fusobacteria* showed an overgrowth with respect to healthy control subjects (mean relative proportion: 23.71% vs. 3.53%, *p* < 0.05; 1.27% vs. 0.12%, *p* = 0.59). The authors did not find an effect of probiotics and enteral nutrition on gut microbiota [48]. A more recent pilot study, carried out on 25 children with sepsis, showed a lower diversity of gut microbiota than healthy control subjects (*p* < 0.001). On the genus level, children with sepsis had more opportunistic pathogens, such as *Acinetobacter* and *Enterococcus*, while fewer beneficial bacteria, such as *Roseburia*, *Bacteroides,* and *Faecalibacterium*, were detected [49]. Lankelma JM and colleagues analyzed the gut microbiota of 34 critically ill patients (divided into 25 septic patients and 9 without septic diagnosis), showing that *Faecalibacterium*, *Blautia*, *Ruminococcus*, *Subdoligranulum,* and *Pseudobutyrivibrio* were the most dominant genera (all *p* < 0.0001, healthy controls vs. ICU patients) [12]. Finally, Liu W. and colleagues analyzed the gut microbiota of two different groups of septic patients, showing a higher abundance of *Bacteroides* in septic patients of the E1 group with respect to the E2 group (*p* < 0.0001, Mann–Whitney–Wilcoxon test; 0.206 ± 0.014) and higher abundance of *Enterococcus* in septic patients of the E2 group with respect to the E1 group (*p* < 0.0001, Mann–Whitney–Wilcoxon test; 0.180 ± 0.033 [50]. A summary of phyla and genera variability in the gut microbiota comparing septic and healthy subjects is shown in Figure 4.

The gut microbiota participates in the development of sepsis and influences the susceptibility of the host to the disease [70]. Furthermore, gut–liver crosstalk may provide a basis for the treatment of sepsis-induced organ damage. Reinforcement of the gut barrier function is a good method to reduce the translocation of bacteria and bacterial metabolites and alleviate sepsis-induced organ damage [71]. For this reason, extensive studies are required, particularly on sepsis, because a better understanding of the gut–liver interaction in sepsis may help prevent and limit sepsis-induced liver damage and improve the prognosis of patients with sepsis. Furthermore, more detailed knowledge of the mechanistic role of the gut microbiota in sepsis will allow the development of original and effective approaches to reduce the morbidity and mortality of sepsis [12].

## 6. Therapeutic Approach

Currently, some available therapeutic strategies are aimed at modulating the composition of the intestinal microbiota during infections or as prophylactic therapy in critically ill patients and in patients prone to recurrent infections. Selective intestinal decontamination (SID) is a prophylactic strategy that consists of the administration of non-absorbable topical antimicrobial agents (that preserve anaerobic microbiota) from the oropharynx and upper gastrointestinal tract, without or in association with a short-term course of intravenous broad-spectrum antibiotics, in order to reduce or prevent the incidence of endogenous infection [72,73]. Another strategy often used is the oral administration of beneficial bacteria or factors that positively influence the intestinal microbiota. These include probiotics (living microorganisms that support the functions of the intestinal microbiota), prebiotics (non-digestible oligosaccharides which have a positive impact on the growth and activity of beneficial intestinal bacteria), and symbiotics (a combination of probiotic and prebiotic agents). This combination of probiotics and prebiotics guarantees longer-term action, and it also supplies an energy substrate for some intestinal species [74]. Finally, the gut microbiota can be modulated via a faecal microbiota transplant (FMT), reducing pathogenic microorganisms through administering a healthy donor’s faecal material into a sick person’s intestine through enema or endoscopically [75]. While antibiotics are one of the greatest inventions of the twentieth century, their potential side effects are being increasingly recognized. In addition to concerns about the emergence of bacterial resistance, it has been suggested that antibiotic-induced changes to intestinal flora could have serious and long-lasting consequences for human physiology [58,59]. Paradoxically, antibiotic treatment can increase susceptibility to opportunistic and nosocomial infections through affecting the resistance of the intestinal microbiota to colonization [76]. Decreased gut microbial diversity could be of relevance for critically ill patients, as the vast majority of patients in an ICU are treated with antibiotics. A prospective point prevalence study involving 1265 ICUs across the world found that on any given day, 75% of patients admitted to these ICUs received antibiotics [77]. Antibiotics, especially those with anti-anaerobic activity, dramatically alter microbial ecology and result in the acquisition and domination by normally low-abundance but highly pathogenic species, such as *Enterococcus faecium* or *Klebsiella pneumoniae*. It has been reported that domination by Enterococcus increased the risk of subsequent VRE bacteremia nine-fold whereas domination by *Proteobacteria* increased the risk of Gram-negative bacteremia five-fold. The association between microbiota disruption and the risk of sepsis has been also reported. Taken together, these studies indicate that normal microbiota are our first line of defense against pathogens and that their disruption can increase the risk of serious life-threatening infection (e.g., bacteremia) and sepsis [52]. Probiotics may restore the composition of the gut microbiota and introduce beneficial functions to gut microbial communities, resulting in the amelioration or prevention of gut inflammation and other intestinal or systemic disease phenotypes [78]. Furthermore, probiotics could reduce pro-inflammatory cytokines such as TNFα and IL-1β and promote the anti-inflammatory role of IL-10 [79]. On the other hand, this study could support the importance of a metagenomic approach to evaluate the gut microbiota composition because the FMT is an innovative direct therapeutic approach. In 2013, FMT was approved by the Food and Drug Administration (FDA) to treat recurrent *Clostridium difficile* infection, and it is also widely investigated in the treatment of cancer, diabetes, multiple sclerosis, atherosclerosis, hypertension, Parkinson’s disease, and hepatic encephalopathy. Recent studies showed that capsule FMT has an overall response rate of more than 90% and is minimally invasive. Although these treatments showed promising results, they were investigated in preclinical models, or the sample sizes were too small [80]. Prebiotics are substances widely used to maintain normal homeostasis of the gut microbiota [81]. Their intake promotes the growth of good bacteria such as *Lactobacillus* and *Bifidobacterium* while inhibiting the growth of pathogens [82]. In addition, a recent meta-analysis conducted on preterm infants showed that administration of prebiotics in the first few weeks of life is effective in reducing the prevalence rates of sepsis and death and shortening the length of hospital stay [83]. In Box 1 are reported take-home messages.

Box 1TAKE-HOME MESSAGESThe microbiota is involved in nutrient metabolism, immune-modulation, and protection of the gastro-intestinal tract.During sepsis, there is an important disruption of the gut microbiota’s composition, characterized by an important reduction in microbial species diversity.Blood culture is the “gold standard” to diagnose sepsis but requires a long time. Biomarkers (CRP, PCT, and Presespin) can be very helpful for an earlier diagnosis.The use of antibiotics is mandatory but leads to a loss of important taxa, alters certain metabolic pathways, and induces microbiota to enter into a state of resilience against pathogens.Selective intestinal decontamination (SID); administration of probiotics, prebiotics, and symbiotics; and faecal microbiota transplants (FMTs) are therapeutic strategies effective in modulating the composition of the intestinal microbiota during infections or as prophylactic therapy.

## 7. Conclusions and Future Perspective

Up-to-date knowledge is needed on the composition of the gut microbiota in critically ill patients to better understand the clinical consequences of microbiota disturbances and, thus, determine which patients might benefit from microbiota-based therapies. Indeed, there are currently therapeutic approaches such as pre- or probiotics and FMT; new strategies are also being clinically studied to target specific microbial components, such as metabolites, phages, and miRNAs.

More research is required to understand which diversity alteration in the microbial intestinal community leads to the onset or the exacerbation of many diseases. However, molecular biology and sequencing techniques on gut microbiota could be used as a screening test to identify high-risk populations, allowing them to prevent such disorders. In confirmed cases of septicemia, an analysis of intestinal flora is of paramount importance due to worsening of dysbiosis during the hospitalization period. In particular, as we discussed in this review, microbiota disorders can lead to organ failure [84]. In the near future, microbiota analysis should be implemented in microbiology laboratories alongside routine diagnosis. Even if NGS technology is a time-consuming process, the SARS-CoV-2 pandemic helped to train healthcare staff and to implement the automation of methodologies such as library preparation [85].

NGS sequencing combined with artificial intelligence is currently being investigated to evaluate gut microbiota as a predictive biomarker for therapeutic response [86]. This approach may help clinicians to improve their clinical experience, opening new perspectives on personalized therapy and increasing the healthy life expectancy of patients [87,88]. In conclusion, the gut microbiota could be a very important bedside tool for diagnosis and could be very helpful to prevent or treat several diseases.

## Figures and Tables

**Figure 1 ijms-24-09307-f001:**
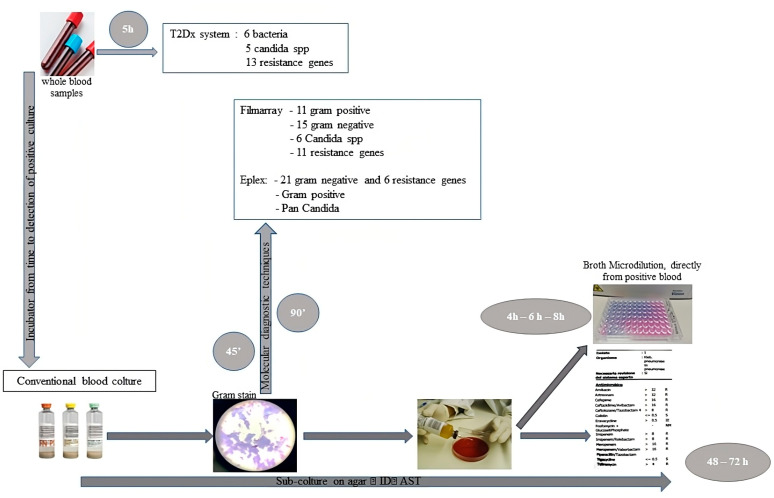
T2Dx system utilizes non-culture T2 MRI technology to detect nucleic acids and microbial cells directly in whole blood samples, for the rapid detection of BSI caused by a modified panel of ESKAPE bacterial species or *Candida* spp. In the conventional method, blood culture (BC) bottles are inserted into automated BC incubators with constant monitoring [29]. In case of growth detection, a Gram stain is prepared from the positive BC broth. At the same time, the positive BC broth is subcultured on agar plates, followed by overnight incubation to grow microbial colonies that are used for identification and antimicrobial susceptibility testing (AST). Through using antibiograms in broth microdilution directly from the sample, the time required to obtain results is drastically reduced [30]. Molecular diagnostic tests based on nucleic acid amplification, usually via PCR, on positive BC bottle specimens allow pathogen identification and detection of some drug resistance markers with high sensitivity and specificity.

**Figure 2 ijms-24-09307-f002:**
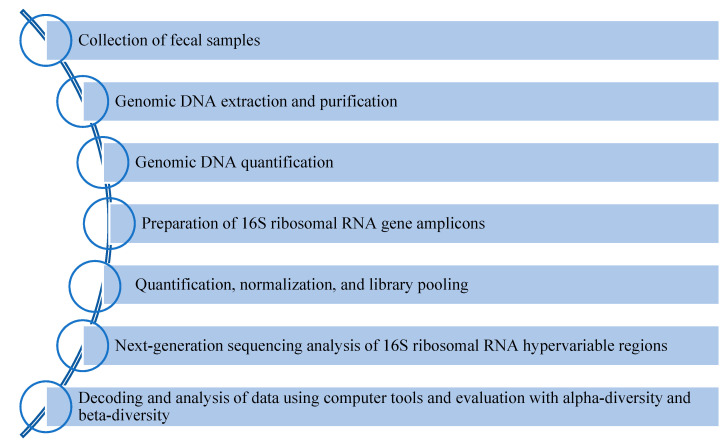
Workflow of gut microbiota analysis.

**Figure 3 ijms-24-09307-f003:**
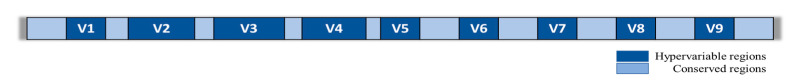
Schematic representation of 16SrRNA conserved (light blue) and hypervariable (deep blue) regions.

**Figure 4 ijms-24-09307-f004:**
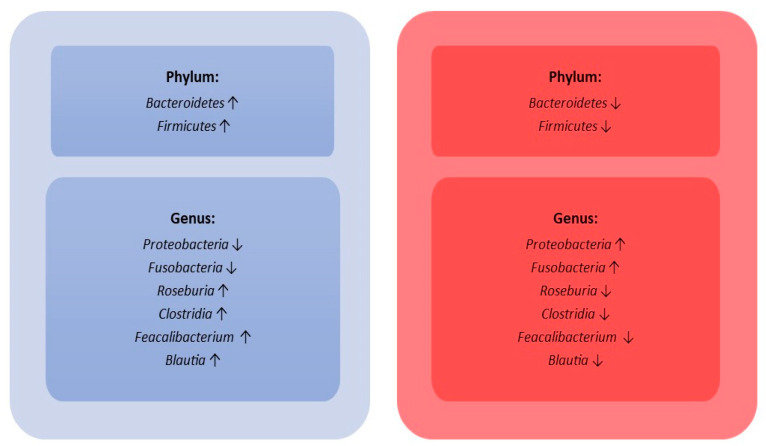
Gut microbiota variability according to healthy status (light blue box) and during sepsis (red box). The arrows indicate the increase or the decrease of phylum and/or genus.

**Table 1 ijms-24-09307-t001:** Gut microbiota analysis via 16SrRNA NGS sequencing: variable region sequenced and platform used.

Sample Size	Variable Region Sequenced	NGS Platform	Sequencing Performance by Taxonomy	References
172 healthy subjects	V1-V3, V3-V4 and V4.	MiSeq (Illumina, CA, USA); PacBio (Pacific Biosciences, CA, USA).	Higher discrimination of *Ruminococcaceae* and *Sphingomonas* (V1-3), *Akkermansia* (V3-4), *Haemophilus*, *Methanobrevibacter,* and *Citrobacter* taxa (V4).	Whon TW. et al., 2018 [44].
33 healthy subjects	V3-V4 and V4-V5.	MiSeq (Illumina, CA, USA).	Higher discrimination of *Actinomyces*, *Alistipes*, *Bacteroides*, *Cellulosimicrobium*, *Parabacteroides,* and *Flavonifractor* genera (V3-V4).	Abellan-Schneyder I. et al., 2021 [45].
192 healthy subjects	V1-V2 and V3-V4.	MiSeq (Illumina, CA, USA).	Higher discrimination of *Bacteroidetes*, *Proteobacteria*, and *Actinobacteria phyla* (V1-V2).	Kameoka S. et al., 2021 [46].
5 healthy subjects	V1-V2, V3-V4 and V4.	MiSeq (Illumina, CA, USA).	Lower discrimination of *Bifidobacteriales* and higher discrimination of *Enterobacteriales* and *Erysipelotrichales* (V1-V2); higher discrimination of *Clostridiales* and lower discrimination of *Bacteroidales*, *Betaproteobacteriales*, *Choriobacteriales,* and *Pasteurellales* (V3-V4); discrimination of *MollicutesRF39* (V4).	Chen Z. et al., 2019 [47].
15 septic shock patients vs. 15 healthy control subjects	V3-V4.	MiSeq (Illumina, CA, USA).	Higher abundance of *Proteobacteria* and *Fusobacteria* in septic shock patients compared to healthy control subjects.	Wan YD et al., 2018 [48].
25 septic children vs. 15 healthy control subjects	V3-V4.	HiSeq (Illumina, CA, USA).	Gut microbiota diversity in septic children lower than healthy control subjects. Higher abundance of *Acinetobacter* and *Enterococcus* and lower abundance of *Roseburia*, *Bacteroides*, *Clostridia*, *Faecalibacterium,* and *Blautia* in septic children compared to healthy control subjects.	Du B. et al., 2021 [49]
34 critically ill patients (25 septic patients and 9 without septic diagnosis) vs. 15 healthy control subjects	V1-V2.	MiSeq (Illumina, CA, USA).	*Firmicutes* and *Bacteroidetes Phyla* constituted <89% of all bacteria. *Faecalibacterium*, *Blautia*, *Ruminococcus*, *Subdoligranulum,* and *Pseudobutyrivibrio* were the most dominant genera.	Lankelma JM et al., 2017 [12]
131 septic patients vs. 264 healthy control subjects (E1 group); 129 septic patients vs. 26 healthy control subjects (E2 group).	V3-V4.	MiSeq (Illumina, CA, USA).	Higher abundance of *Bacteroides* in septic patients of E1 group with respect to E2 group and higher abundance of *Enterococcus* in septic patients of E2 group with respect to E1 group.	Liu W. et al., 2020 [50]

**Table 2 ijms-24-09307-t002:** Recognized effect of antibiotics on gut microbiota.

Antibiotics	Microbiota Composition		
	Gram-Positive	Gram-Negative	Anaerobes
Cetriaxone	increase	largest decrease	no changes
Amoxicillin	increase	increase	no changes
Clindamycin	increase	increase	largest decrease
Fluoroquinolones	no changes	largest decrease	no changes
Metronidazole	no changes	no changes	no changes
Macrolide	decrease	increase	decrease
Vancomycin	increase or decrease	no changes	decrease

## Data Availability

Not applicable.

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
