# Peer review of "The Role of Gut Microbiota in the Clinical Outcome of Septic Patients: State of the Art and Future Perspectives"

_ijms, 2023, doi:10.3390/ijms24119307_

Round 1

Reviewer 1 Report

In the review Marascio et al provide a comprehensive overview of the microbiome and it's association with sepsis. The authors provide a thorough review of the host response to sepsis and following an introduction to the microbiome provide a very thorough review of 16S rRNA sequencing strategies. The authors discuss in detail markers for diagnosis of sepsis and the interventions that are currently used. The authors provide detail on the use of antimicrobials and the effects of broad spectrum antibiotics on the gut microbiome and how the imbalances of  the microbiome as a consequence of antimicrobial administration could leave to further adverse events. Alternative therapeutic strategies are discussed including the use of probiotics and FMT to maintain the healthy microbiome. Overall the review is well structured and reads well.

Line 283: Please change "Authors not found" to "The authors did not find  an effect...."

Line 313: Please change to " Currently , some therapeutic strategies are available....

Author Response

Comments and Suggestions for Authors

In the review Marascio et al provide a comprehensive overview of the microbiome and it's association with sepsis. The authors provide a thorough review of the host response to sepsis and following an introduction to the microbiome provide a very thorough review of 16S rRNA sequencing strategies. The authors discuss in detail markers for diagnosis of sepsis and the interventions that are currently used. The authors provide detail on the use of antimicrobials and the effects of broad spectrum antibiotics on the gut microbiome and how the imbalances of the microbiome as a consequence of antimicrobial administration could leave to further adverse events. Alternative therapeutic strategies are discussed including the use of probiotics and FMT to maintain the healthy microbiome. Overall the review is well structured and reads well.

Dear Reviewer#1,

Thank you so much for your comments on our work. We are glad that you appreciated the manuscript from a scientific point of view. According to your suggestion, we made changes highlighted with “Track Changes” function using MS Word.

Comments on the Quality of English Language

Line 283: Please change "Authors not found" to "The authors did not find an effect...."

Reply 1: We wrote "The authors did not find an effect...." instead of "Authors not found" at line 283.

Line 313: Please change to " Currently, some therapeutic strategies are available....

Reply 2: We wrote "Currently, some therapeutic strategies are available....” at line 313.

Reviewer 2 Report

The work is well structured, written, but before being considered for publication, minor corrections are necessary, which I will detail in the following:

Lines 26 and 30: I recommend inserting more references

Lines 46-48: insert references

Lines 59-61: insert references

Lines 61-63: insert references

Lines 75-77: insert references

Lines 81-86: it is a personal finding or the information is taken from specialized literature. In this case, references are needed to support the statements

Lines 91-94: insert references

The reformulation of the conclusions chapter, because I believe that the importance of this good microbiota and its role in the evolution of septicemia in confirmed cases should be detailed more. Also, how the authors see the implementation of molecular biology methods in the diagnosis of septicemia, respectively in the establishment of microbiota, in the near future in microbiological diagnostic laboratories.

Author Response

Comments and Suggestions for Authors

The work is well structured, written, but before being considered for publication, minor corrections are necessary, which I will detail in the following:

Dear Reviewer#2,

Thank you very much for your comments to improve the quality of manuscript. We are glad that you appreciated the manuscript from a scientific point of view. According to your suggestion, we made changes highlighted with “Track Changes” function using MS Word.

Lines 26 and 30: I recommend inserting more references.

Reply: according to your suggestion, we added the references [3] Thursby, E., Juge, N. Introduction to the human gut microbiota. Biochem J. 2017, 474, 1823, 1836. And [4] Nova, E., Gómez-Martinez, S., González-Soltero, R. The Influence of Dietary Factors on the Gut Microbiota. Microorganisms. 2022, 10, 1368.

Lines 46-48: insert references.

Reply: according to your suggestion, we added the reference [14] Huang, M., Cai, S., Su, J. The Pathogenesis of Sepsis and Potential Therapeutic Targets. Int J Mol Sci. 2019, 20, 5376.

Lines 59-61: insert references.

Reply: according to your suggestion, we added the reference [16] Li, A. T., Moussa, A., Gus, E., Paul, E., Yii, E., Romero, L., Lin, Z. C., Padiglione, A., Lo, C. H., Cleland, H., et al. Biomarkers for the Early Diagnosis of Sepsis in Burns: Systematic Review and Meta-analysis. Ann Surg. 2022, 275, 654, 662.

Lines 61-63: insert references.

Reply: according to your suggestion, we added the reference [17] Hotchkiss, R. S., Moldawer, L. L., Opal, S. M., Reinhart, K., Turnbull, I. R., Vincent, J. L. Sepsis and septic shock. Nat Rev Dis Primers. 2016 2, 16045.

Lines 75-77: insert references.

Reply: according to your suggestion, we added the references [18] Serapide, F.; Quirino, A.; Scaglione, V.; Morrone, H.L.; Longhini, F.; Bruni, A.; Garofalo, E.; Matera, G.; Marascio, N.; Scarlata, G.G.M., et al. Is the Pendulum of Antimicrobial Drug Resistance Swinging Back after COVID-19? Microorganisms 2022, 10, 957. And [19] Bongiorno, D., Bivona, D. A., Cicino, C., Trecarichi, E. M., Russo, A., Marascio, N., Mezzatesta, M. L., Musso, N., Privitera, G. F., Quirino, A., et al. Omic insights into various ceftazidime-avibactam-resistant Klebsiella pneumoniae isolates from two southern Italian regions. Front Cell Infect Microbiol. 2023, 12, 1010979.

Lines 81-86: it is a personal finding or the information is taken from specialized literature. In this case, references are needed to support the statements.

Reply: the statements were not a personal finding, we added the following references [20] Wang, M.; Liu, H.; Ren, J.; Huang, Y.; Deng, Y.; Liu, Y.; Chen, Z.; Chow, F.W.-N.; Leung, P.H.-M.; et al. Enzyme-Assisted Nucleic Acid Amplification in Molecular Diagnosis: A Review. Biosensors 2023, 13, 160.

Lines 91-94: insert references.

Reply: according to your suggestion, we added the references [21] Bassetti, M., Russo, A., Righi, E., Dolso, E., Merelli, M., Cannarsa, N., D'Aurizio, F., Sartor, A., Curcio, F. Comparison between procalcitonin and C-reactive protein to predict blood culture results in ICU patients. Crit Care. 2018, 22, 252. [22] Bassetti, M., Russo, A., Righi, E., Dolso, E., Merelli, M., D'Aurizio, F., Sartor, A., Curcio, F. Role of procalcitonin in bacteremic patients and its potential use in predicting infection etiology. Expert Rev Anti Infect Ther. 2019, 17, 99, 105. [23] Bassetti, M., Russo, A., Righi, E., Dolso, E., Merelli, M., D'Aurizio, F., Sartor, A., Curcio, F. Role of procalcitonin in predicting etiology in bacteremic patients: Report from a large single-center experience. J Infect Public Health. 2020, 13, 40, 45. [24] Russo, A., Venditti, M., Ceccarelli, G., Mastroianni, C. M., d'Ettorre, G. Procalcitonin in daily clinical practice: an evergreen tool also during a pandemic. Intern Emerg Med. 2021, 16, 541, 543. [25] Alessandri, F., Pugliese, F., Angeletti, S., Ciccozzi, M., Russo, A., Mastroianni, C. M., d'Ettorre, G., Venditti, M., Ceccarelli, G. Procalcitonin in the Assessment of Ventilator Associated Pneumonia: A Systematic Review. Adv Exp Med Biol. 2021, 1323, 103, 114.

The reformulation of the conclusions chapter, because I believe that the importance of this good microbiota and its role in the evolution of septicemia in confirmed cases should be detailed more. Also, how the authors see the implementation of molecular biology methods in the diagnosis of septicemia, respectively in the establishment of microbiota, in the near future in microbiological diagnostic laboratories.

Reply: Thank you very much for your suggestion. To improve this section, we added some sentences and related references according with your suggestions.